# Short-Term Solar Irradiance Forecasting Based on a Hybrid Deep Learning Methodology

**Ke Yan [1]**, **Hengle Shen [1]**, **Lei Wang [2,*]**, **Huiming Zhou [3]**, **Meiling Xu [4]** and **Yuchang Mo [5]**

1    Key Laboratory of Electromagnetic Wave Information Technology and Metrology of Zhejiang Province, College of Information Engineering, China Jiliang University, Hangzhou 310018, China; yanke@cjlu.edu.cn (K.Y.); p1803085222@cjlu.edu.cn (H.S.)
2    School of Computer Engineering, Weifang University, Weifang 261061, China
3    Zhejiang Huayun Information Technology Co., Ltd., Hangzhou 310008, China; zhouhuiming@hyit.com.cn
4    Nanhu College, Jiaxing University, Jiaxing 314001, China; meilingxunh@gmail.com
5    Fujian Province University Key Laboratory of Computational Science, School of Mathematical Sciences, Huaqiao University, Quanzhou 362021, China; myc@hqu.edu.cn
*    Correspondence: wanglandpqpq@hotmail.com; Tel.: +86-53-6878-5510

**Abstract:** Accurate prediction of solar irradiance is beneficial in reducing energy waste associated with photovoltaic power plants, preventing system damage caused by the severe fluctuation of solar irradiance, and stationarizing the power output integration between different power grids. Considering the randomness and multiple dimension of weather data, a hybrid deep learning model that combines a gated recurrent unit (GRU) neural network and an attention mechanism is proposed forecasting the solar irradiance changes in four different seasons. In the first step, the Inception neural network and ResNet are designed to extract features from the original dataset. Secondly, the extracted features are inputted into the recurrent neural network (RNN) network for model training. Experimental results show that the proposed hybrid deep learning model accurately predicts solar irradiance changes in a short-term manner. In addition, the forecasting performance of the model is better than traditional deep learning models (such as long short term memory and GRU).

**Keywords:** short-term forecasting; solar irradiance; gated recurrent unit; attention mechanism

## 1. Introduction

While the overall world's energy consumption increases, photovoltaic energy generation is becoming increasingly important. Non-renewable energy resources (coal, oil, natural gas, etc.) have the disadvantages of having limited storage, providing high pollution, and causing landscape changes [1]. As part of the process of replacing traditional energy resources with renewable energy resources, the factor of environmental protection is highly relevant. As a result, clean and non-polluting renewable energy resources (solar, wind, and geothermal, etc.) have been attracting both scientists' and engineers' attention [2]. Moreover, for countries with a large landscape, such as China, the use of solar energy as a replacement for traditional oil-based energy resources is an urgent priority. The solar radiation in the whole country is 3340 MJ/m$^2$–8400 MJ/m$^2$. An efficient and effective way of utilizing solar irradiance power is highly demanded for those countries [3,4].

Solar irradiance forecasting has been widely studied in related fields [5]. Accurate forecasting of solar irradiance provides important evidence for predicting photovoltaic energy generation at the same location, since solar irradiance is directly proportional to photovoltaic energy generation using solar panels. However, solar irradiance prediction is challenging because it is highly co-related to various environmental factors, such as the sun position, temperature, wind speed, and cloud movement. Accurately predicting solar irradiance is a complex and difficult task.

Machine learning and deep learning technologies are popular data-driven prediction models for time series data forecasting [6,7], including artificial neural networks [8], grey theory methods [9], and support vector regression [10]. There are two main types of research on hybrid learning forecasting methods: (1) serialized ensemble learning methods, which combine two or more prediction models, such as the clustering method and least squares support vector machine algorithm [10], a combination of wavelet transform and neural network [11], and so on. (2) parallel ensemble learning methods, which combine the prediction results obtained by different prediction methods, where different prediction weights are given to each prediction method based on certain optimization criteria to form a final prediction result. Before this study, many different machine learning methods have been proposed before to predict the power output of PV systems, such as the BP Neural Network (BPNN) [11], Radial Basis Function Neural Network (RBFNN) [12], generative adversarial network (GAN) [13], semi-supervised algorithm [14], fuzzy inference method [15], mathematical analysis methods, and so on.

In this study, a hybrid deep learning framework has been proposed to forecast the solar irradiance changes at the Nevada desert in the USA. The solar irradiance is directly proportional to the energy outputs of photovoltaic plants at the Nevada desert. Multi-dimensional data inputs for the deep learning neural network were adopted. Additional input features include the average wind speed and peak wind speed, which make the prediction results more accurate and reliable. The experiment results also demonstrate that the solar irradiance prediction results considering multiple factors (such as the wind speed) are significantly better than the prediction result of a single factor. This research has greatly improved the prediction accuracy of the solar irradiance, providing a basis for the energy generation forecasting of photovoltaic systems.

## 2. Materials and Methods

### 2.1. Introduction to the Comparison Model

In this paper, a series of models are used to predict the short-term solar irradiance of the same dataset. The models used are briefly described as follows:

The Long short term memory (LSTM) model [16]: the LSTM model uses two LSTM layers to extract important features of the input power generation time series and data sets, respectively. It then extracts the extracted feature vectors into one-dimensional vectors and fuses them, and then outputs the prediction results through a fully connected layer.

The Gated recurrent unit (GRU) model: unlike LSTM, GRU has a simple structure with only two gates: a reset gate and an update gate. Its biggest advantages are that it requires fewer parameters, has a fast training speed, and saves time.

### 2.2. Gated Recurrent Unit (GRU) with an Attention Mechanism

#### 2.2.1. Inception Network

The usual way to improve network capabilities is to increase the depth and width of the network, but this also provides a lot of "side effects", such as model overfitting, gradient disappearance, and so on. The Inception Network provides a solution to this problem, so that the amount of calculation does not need to change, and the depth and width of the network are improved.

The Inception Network [17] has gone through multiple versions from InceptionV1 (also known as "GoogLeNet") to InceptionV3. In the improvement to InceptionV2, the convolution of two $3 \times 3$ convolution was removed in favor of $5 \times 5$, and some minor improvements of Batch Normalization were proposed. When it was improved to InceptionV3, the main changes involved the decomposition of the convolution kernel size: this convolution kernel is now decomposed into a symmetric small convolution kernel or an asymmetric convolution kernel. The network structure of InceptionV3 is shown in Figure 1.

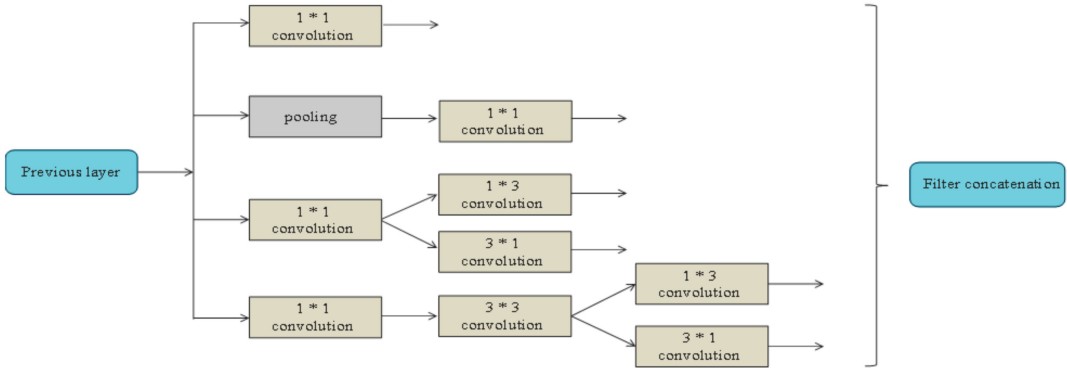

**Figure 1.** The inceptionV3 network structure.

### 2.2.2. ResNet Network

After solving the problem of increasing network depth, theoretically, more complex features can be extracted to achieve better results. But in practice, degradation problems occur. This means the network accuracy increases while the network depth increases. Subsequently, there is a trend of saturation or even decline in network depth, meaning the difficulty of model training increases, which can be further revolved using the ResNet network structure [18].

The main structural residuals of the ResNet network enable the ResNet network to use multiple referenced layers to learn the residual representation between input and output, instead of using a reference layer to learn the input-to-output mapping like in a traditional network. The output of the residual unit is added to by the concatenated output of the multiple convolutional layers and the input elements (guaranteeing that the convolutional layer output and the input element dimensions are the same), and then the activation of ReLU, the structure is cascaded to obtain the residual network, which is the ResNet network. Experiments show that the residual structure can effectively accelerate the training convergence speed of the model, and can also effectively mitigate the degradation problem. Its residual structure is shown in Figure 2.

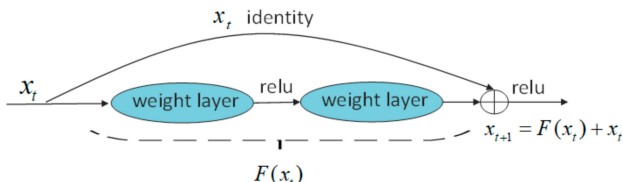

**Figure 2.** The ResNet network residual structure.

This experiment combined Inception and Residual "strong combination" into a new Inception_ResNet structure. The specific structure is shown in Figure 3:

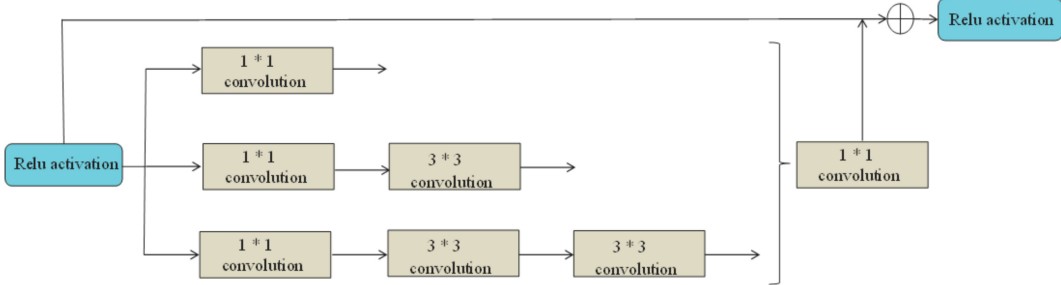

**Figure 3.** Inception_ResNet structure.

### 2.2.3. GRU-Gated Cyclic Neural Network

GRU (Gate Recurrent Unit) [19] is a variant of the Recurrent Neural Network (RNN), which is the same as the LSTM (Long-Short Term Memory) in terms of being used in order to solve the problem of long-term memory network and backpropagation. GRU and LSTM are almost the same in performance, but this paper chose GRU, mainly because GRU is easier to train with LSTM and improves training efficiency substantially. In addition, the model structure is simpler.

Unlike the LSTM, the GRU replaces the forget gate and the input gate in the LSTM with an update gate that controls the cell state information from the previous moment and brings it into the current cell. The larger the value of the update gate is, the more the cell state information can be brought in from the previous time. In addition, a reset gate is used to control how much information is written to the current state from the previous state. In the cell state, the smaller the reset gate is, the less information about the previous cell state is written. At the same time, the GRU also mixes the cell state and the hidden state to achieve simplification of the model. Its forward calculation process can be expressed as:

$$z_t = \sigma(W_z \cdot [h_{t-1}, x_t]), \tag{1}$$

$$r_t = \sigma(W_r \cdot [h_{t-1}, x_t]), \tag{2}$$

$$\widetilde{h_t} = \tanh(W \cdot [r_t * h_{t-1}, x_t]), \tag{3}$$

$$h_t = (1 - z_t) * h_{t-1} + z_t * \widetilde{h_t}, \tag{4}$$

$$y_t = \sigma(W_o \cdot h_t), \tag{5}$$

where [ ] indicates that two vectors are connected, and * indicates the multiplication of the matrices. $z_t$ and $r_t$ respectively represent the output of the update gate and the reset gate, while $W_z$, $W_r$, $W$, and $W_o$ respectively represent the matrix of the corresponding weight coefficient and the offset term. $\sigma$ and tanh respectively represent the sigmoid and the hyperbolic tangent activation function, and $x_t$ is the network input at time $t$. $h_t$ and $h_{t-1}$ represent the hidden layer information of the current time and the previous time, and $\widetilde{h_t}$ represents the candidate state of the input. First, the network input $x_t$ at the time of the hidden state $h_{t-1}$ and $t$ at the last moment calculates the output of the reset gate and the update gate by the Formulae (1) and (2). Then, after the reset of the reset gate $r_t$ determines how much memory is retained, the implicit layer $\widetilde{h_t}$ is calculated by Formula (3), that is, the new information at the current time $t$. Afterwards, by using Formula (4), the update gate $z_t$ determines how much information was discarded at the previous moment, and how much information is retained in the candidate hidden layer $\widetilde{h_t}$ at the current moment. The hidden layer information $h_t$ is then added. Finally, the output of the GRU is passed to the next GRU gated loop unit by using Formula (5): which is a sigmoid function. The internal structure of the GRU gated loop unit is shown in Figure 4.

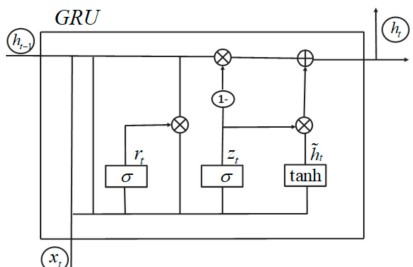

**Figure 4.** A gated recurrent unit (GRU) gated loop unit internal structure.

### 2.2.4. The Attention Mechanism

The attention mechanism [5] was developed based on the human visual attention mechanism. In short, it is a series of attention distribution coefficients, i.e., a series of weight parameters. The attention

mechanism extracts important information without increasing the computation time, making the model's learning process more flexible. In this paper, Attention (the attention mechanism) is added after GRU, and its output vector is used as the input of attention. Attention finds the attention weight by itself, which makes the model's prediction accuracy more accurate. Its calculation equations are as follows:

$$\beta_i = \frac{\exp(e_i)}{\sum\limits_{i=1}^{t} \exp(e_i)}, \sum_{i=1}^{t} \beta_i = 1, \tag{6}$$

$$e_i = \tanh(W_h h_i + b_h), e_i \in [-1, 1]. \tag{7}$$

Attention searches for $h_t$'s attention weight $\beta_t$, where $W_h$ and $b_h$ are weight coefficients and biases. The attention vector $A' = \left\{ h'_1, h'_2, \dots, h'_t \right\}$ is obtained by multiplying the attention weights $\beta_t$ and $h_i$. The specific calculation equation is as shown in Equation (8):

$$A_i' = \beta_i \cdot h_i. \tag{8}$$

### 2.3. Model Structure

The GRU_attention solar photovoltaic power generation prediction model is generally composed of the feature extraction of a CNN network and the regression prediction of a RNN cyclic neural network. Inception_ResNet is the combination of Inception and ResNet, which not only increases the width and depth of the network, but also effectively improves the expressiveness of the network. Specifically, in Inception_ResNet, the Residual structure can not only greatly accelerate the convergence of the Inception network, but also increase the Inception structure and network depth, and ultimately provides the model with higher network accuracy. The basic flow of the GRU_attention model is shown in Figure 5.

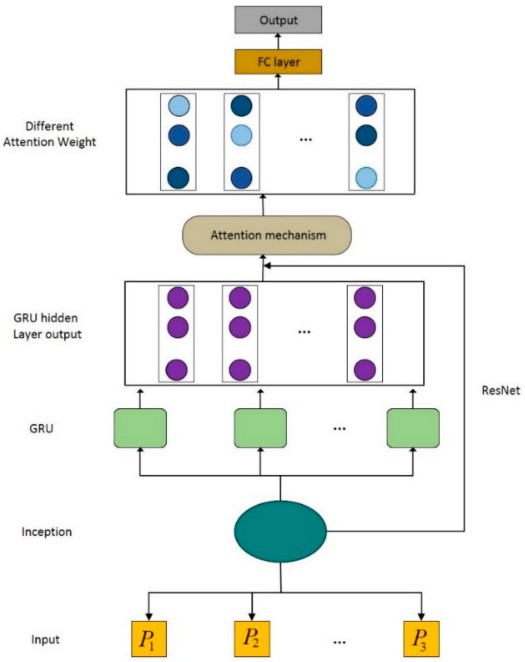

**Figure 5.** GRU_attention basic flow chart.

The overall structure of the GRU_attention model is shown in Figure 6. After the multi-dimensional photovoltaic data is input into the model, the data is first stitched into the form of n × n through the data processing module, and then it is input to the Inception_ResNet network for feature extraction, and then the extracted features are input to the GRU network for training and prediction. The main role of the

data processing part is to transform the data into a specific dimension. The CNN convolutional neural network part uses two Inception structures, namely, Inception_ResNet and InceptionV3 networks, to achieve feature extraction at different scales. The subsequent GRU part uses a two-layer structure to make predictions. Afterwards, the attention mechanism processes the output of the GRU hidden layer, where each element has a different attention weight, and then passes through a Dropout layer to randomly discard hidden neurons. The Dropout operation can effectively improve the model. The training convergence speed can prevent the model from overfitting to a certain extent. Finally, a layer of a fully connected neural network is used to output the prediction results. The entire process is an end-to-end learning model.

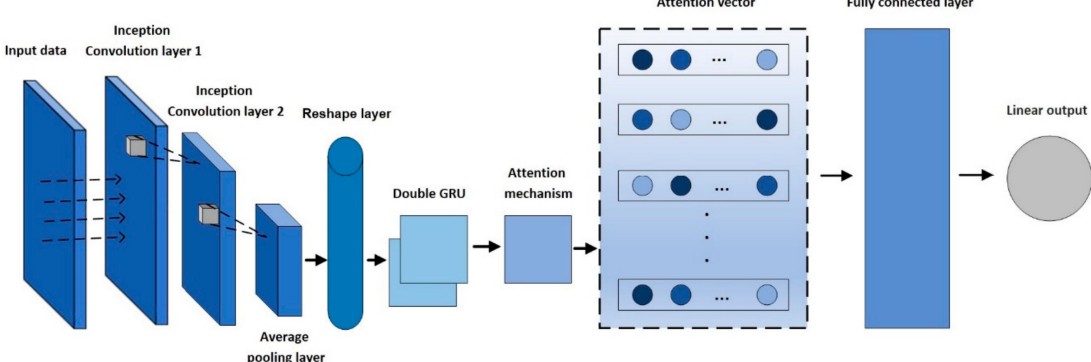

**Figure 6.** GRU_attention model overall structure.

## 2.4. Data Preprocessing

The data used in this study was collected by University of Nevada-Las Vegas and NREL to support solar power generation research in the United States [20]. Local solar irradiance data in the year 2014 was collected. The measurement station is located at latitude: 36.107° north and longitude 115.1425° west. There is missing data in the original data, so the data needs to be pre-processed. For data with missing values, the data at the previous moment is selected for completion. Additional features, such as averaged wind speed and peak wind speed, were used to enhance the prediction performance [21,22]. The data of the whole year 2014 was adopted. For each season, one month of data was selected as the testing dataset and the remaining two months were treated as a training dataset. As a result, we have eight months of solar irradiance data for training and four months of data for testing. Finally, the lagging value was used for the prediction, as shown in Equation (9):

$$\Delta y_x = y_{x+1} - y_x. \tag{9}$$

The Adam network optimizer was selected with a learning rate of 0.002, optimizing the neural network weights of the entire GRU_attention model to minimize the loss value of the loss function.

## 2.5. Experimental Simulation Platform

Hardware configuration: Intel Core (TM) i7-8700K CPU @ 3.70 GHz, NVIDIA, GeForce GTX1080 graphics card, 16GB memory, 8GB video memory.

The software environment was: Python 3.7 (64-bit), tensorflow-gpu version 1.8.0 and keras version 2.0.3.

## 2.6. Evaluation Criteria

Compared with the cyclic architecture, the calculation of the convolution model could be fully parallelized, which can achieve high training efficiency, and at the same time, makes optimization easier. *MAE*, *MAPE,* and *RMSE* were used to evaluate the performance of the prediction model.

The calculation equations of the evaluation criteria *MAE*, *RMSE*, and *MAPE* are shown in Formulae (10)–(12):

$$MAE = \frac{1}{n} \sum_{i=1}^{n} \left| y_i' - y_i \right|, \tag{10}$$

$$RMSE = \sqrt{\frac{1}{n} \sum_{i=1}^{n} (y_i - y_i')^2}, \tag{11}$$

$$MAPE = \frac{1}{n} \sum_{i=1}^{n} \left| \frac{y_i' - y_i}{y_i} \right| \times 100\%, \tag{12}$$

where: $y_i'$ is the result of the model prediction; $y_i$ is the actual test sample value; and $n$ is the total number of test samples. The smaller the values of *MAE*, *RMSE,* and *MAPE*, the better the prediction performance of the model.

## 3. Results

This experiment divides a year of data set into four parts: spring, summer, autumn, and winter. The irradiance pattern in the four seasons changes greatly, which provides the main difficulties for the experiment. This study divided the experiment into 5, 10, 20 and 30 min time intervals to learn the changing trend of the prediction results of solar photovoltaic power generation under three models: LSTM, GRU, and the model GRU_attention proposed in this paper. Table 1 shows the specific evaluation index data of *MAE*, *RMSE*, and *MAPE* for the three models in the four different seasons. The experimental results are shown in Figures 7–10.

**Table 1.** Comparison of the evaluation indexes of each model in the four seasons when the forecast time interval is 5 min, 10 min, 20 min, and 30 min.

|  |  | Spring | | | Summer | | | Autumn | | | Winter | | |
|---|---|---|---|---|---|---|---|---|---|---|---|---|---|
|  |  | *MAE* | *RMSE* | *MAPE* | *MAE* | *RMSE* | *MAPE* | *MAE* | *RMSE* | *MAPE* | *MAE* | *RMSE* | *MAPE* |
|  | LSTM | 26.95 | 36.67 | 6.01 | 59.20 | 89.91 | 9.46 | 13.13 | 18.85 | 7.01 | 21.58 | 44.24 | 9.10 |
| 5 min | GRU | 27.18 | 36.82 | 6.13 | 59.70 | 89.77 | 9.63 | 16.03 | 20.75 | 10.34 | 23.60 | 43.66 | 10.05 |
|  | GRU_Attention | 26.49 | 36.23 | 5.80 | 57.76 | 88.05 | 9.19 | 11.32 | 17.41 | 6.33 | 20.69 | 43.09 | 8.46 |
|  | LSTM | 29.65 | 41.02 | 11.08 | 32.96 | 42.23 | 11.11 | 33.62 | 53.01 | 8.52 | 11.09 | 14.20 | 11.43 |
| 10 min | GRU | 34.42 | 44.71 | 15.04 | 30.92 | 41.08 | 12.66 | 38.35 | 55.00 | 9.92 | 12.83 | 15.20 | 13.21 |
|  | GRU_Attention | 28.44 | 39.72 | 10.29 | 25.52 | 38.82 | 10.44 | 27.28 | 49.49 | 7.85 | 9.48 | 11.44 | 11.89 |
|  | LSTM | 49.09 | 56.22 | 53.85 | 40.58 | 46.31 | 58.06 | 28.11 | 33.86 | 33.55 | 39.10 | 43.54 | 29.38 |
| 20 min | GRU | 36.78 | 45.23 | 49.33 | 47.44 | 53.97 | 61.41 | 24.55 | 29.58 | 29.14 | 37.09 | 41.03 | 26.56 |
|  | GRU_Attention | 21.64 | 27.17 | 22.50 | 30.00 | 36.72 | 21.71 | 14.95 | 20.22 | 18.48 | 28.37 | 34.77 | 16.87 |
|  | LSTM | 47.54 | 58.77 | 45.57 | 47.82 | 58.00 | 50.60 | 59.08 | 81.75 | 47.43 | 52.29 | 61.68 | 48.61 |
| 30 min | GRU | 49.65 | 60.42 | 49.52 | 50.52 | 55.29 | 42.39 | 60.71 | 82.12 | 40.22 | 54.13 | 62.33 | 52.12 |
|  | GRU_Attention | 33.20 | 41.85 | 26.96 | 38.33 | 44.15 | 28.31 | 56.33 | 84.35 | 27.05 | 33.66 | 41.69 | 28.96 |

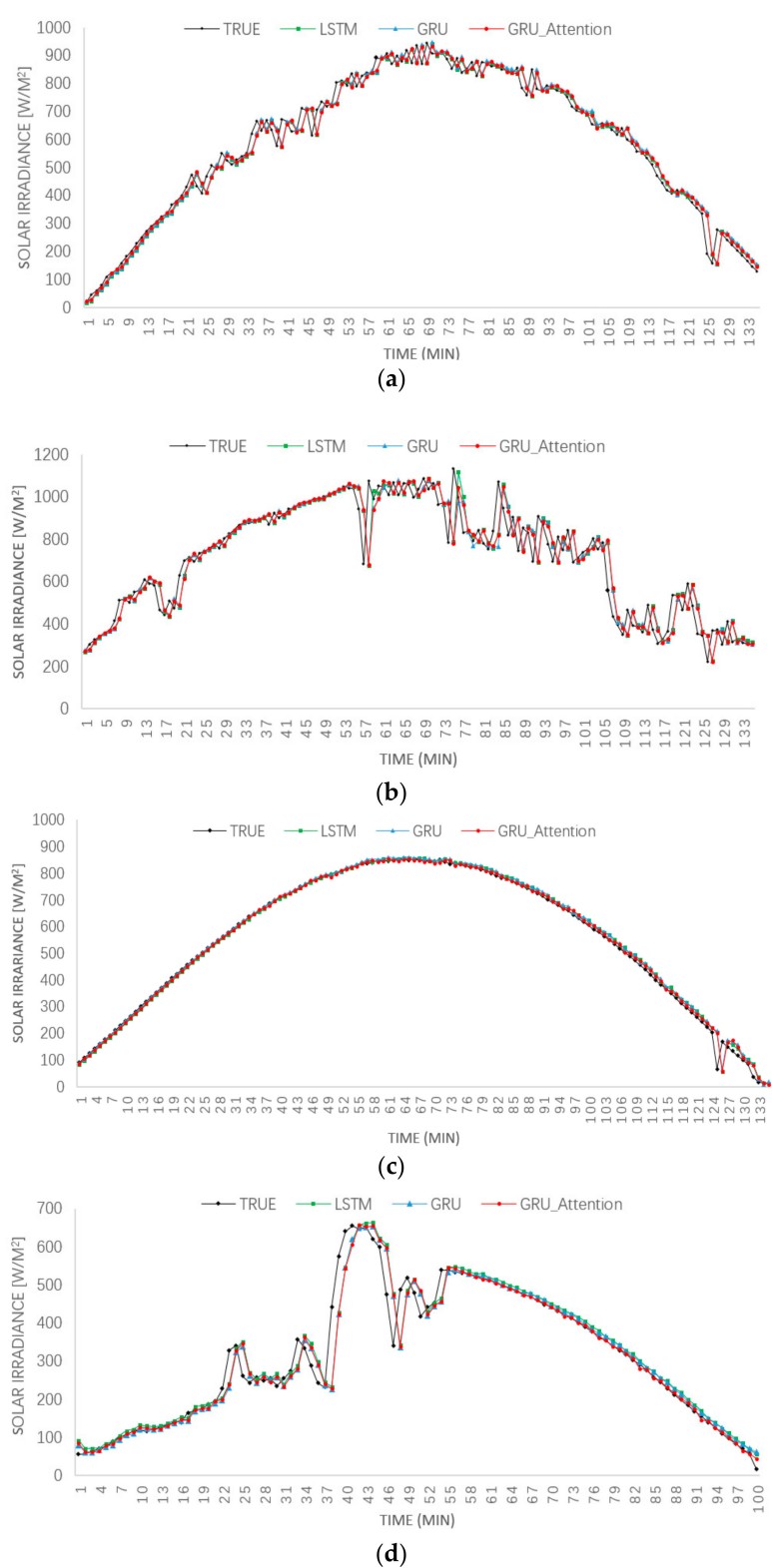

**Figure 7.** The experimental results of each model in which the photovoltaic irradiance prediction time interval is 5 min in 2014. (**a**) spring; (**b**) summer; (**c**) autumn; (**d**) winter.

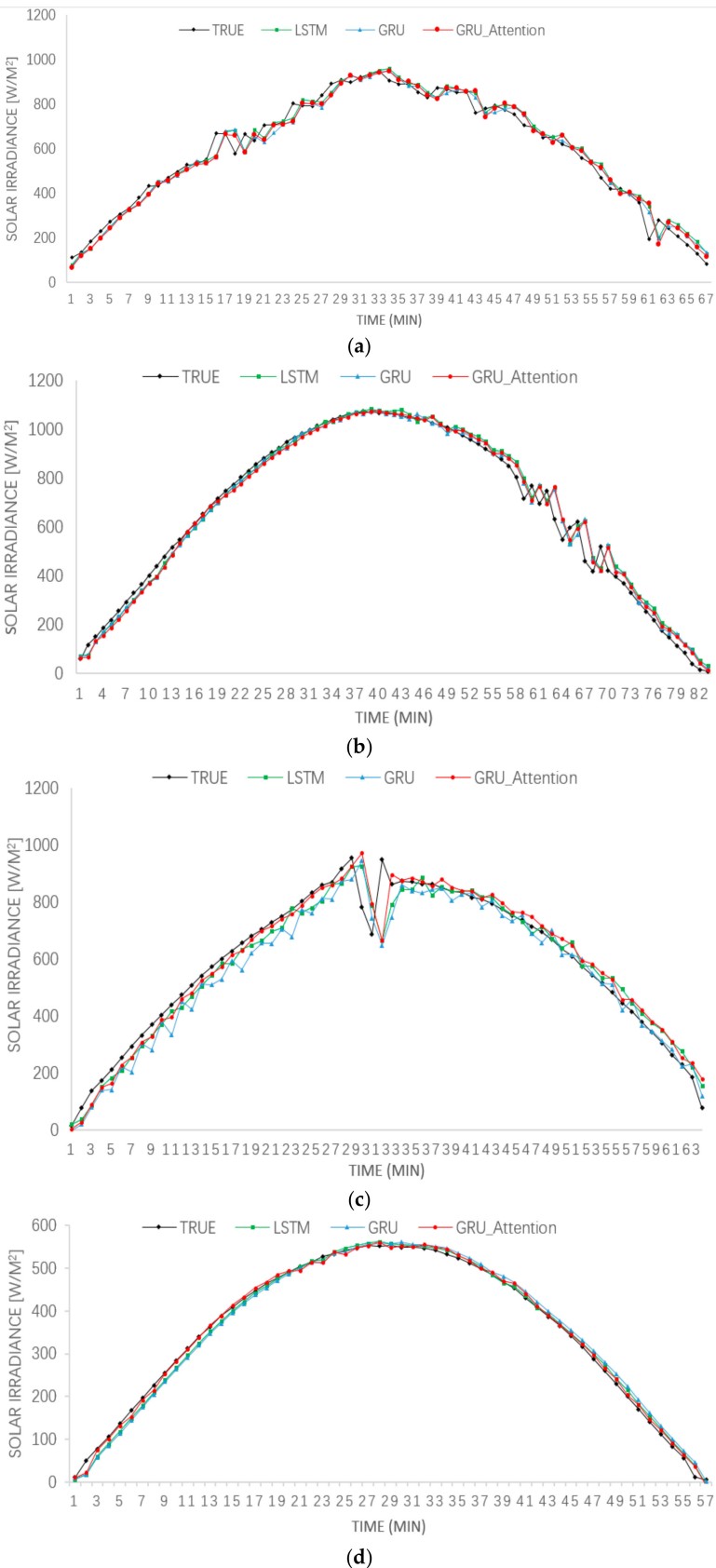

**Figure 8.** The experimental results of each model in which the photovoltaic irradiance prediction time interval is 10 min in 2014. (**a**) spring; (**b**) summer; (**c**) autumn; (**d**) winter.

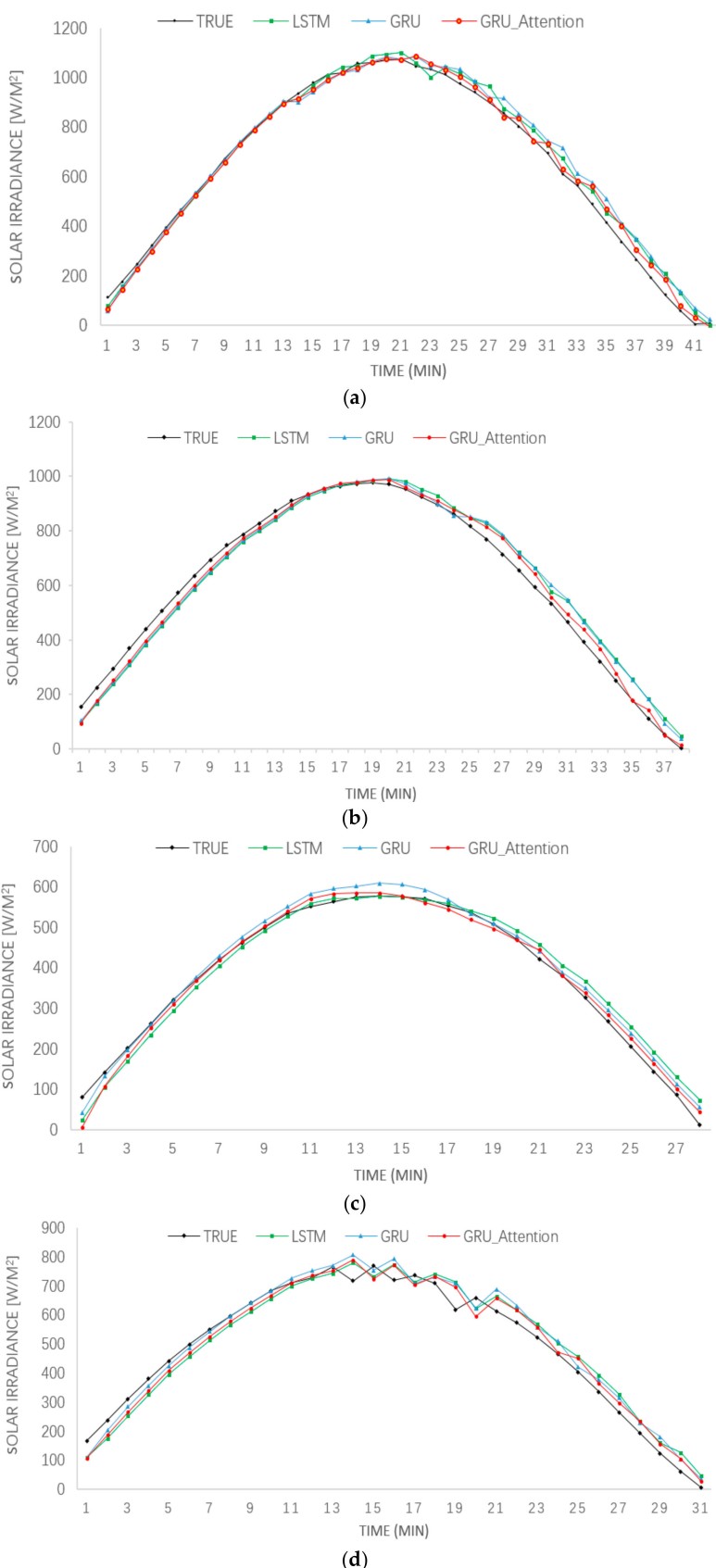

**Figure 9.** The experimental results of each model in which the photovoltaic irradiance prediction time interval is 20 min in 2014. (**a**) spring; (**b**) summer; (**c**) autumn; (**d**) winter.

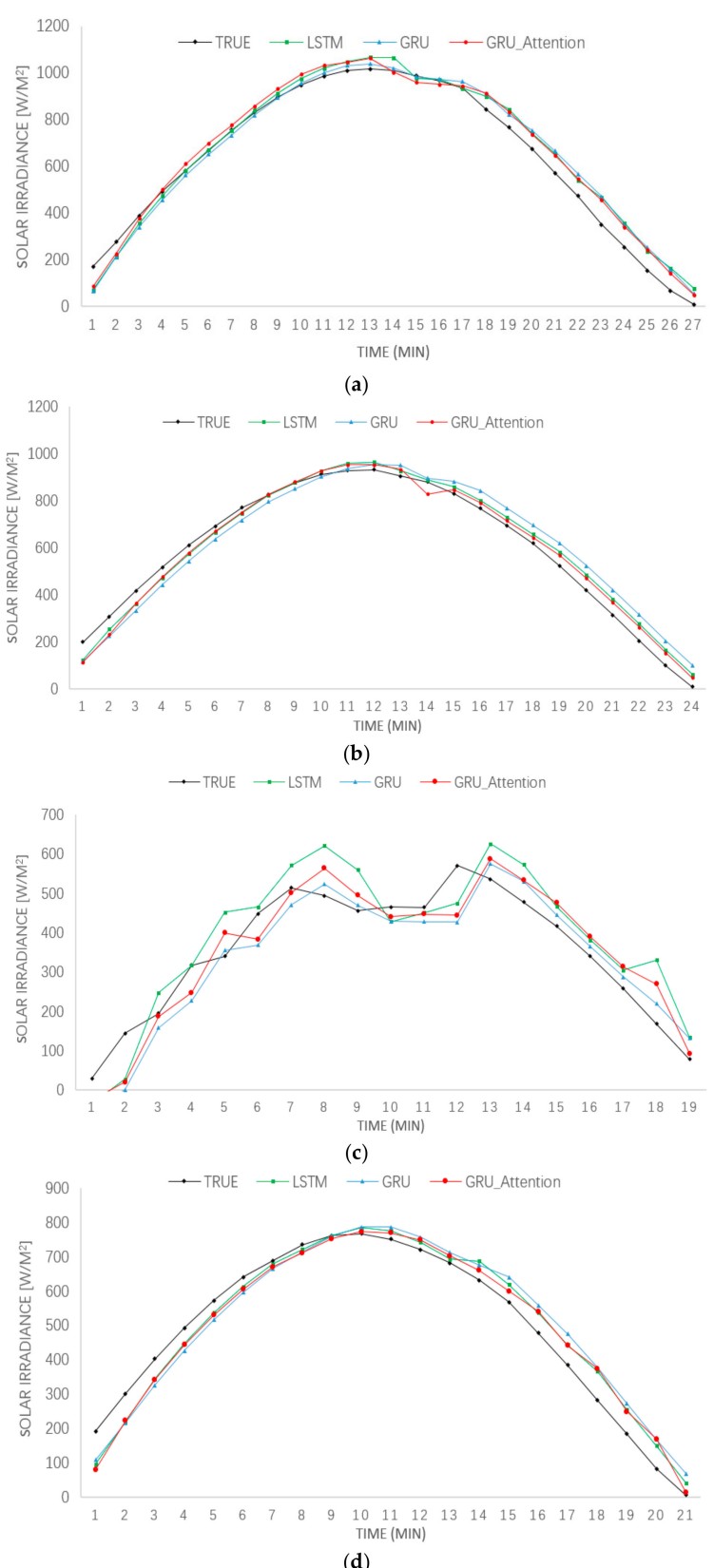

**Figure 10.** The experimental results of each model in which the photovoltaic irradiance prediction time interval is 30 min in 2014. (**a**) spring; (**b**) summer; (**c**) autumn; (**d**) winter.

In order to study the performance of the proposed model in different time ranges, experiments have been conducted in the time ranges of 5 min, 10 min, 20 min, and 30 min, respectively, to predict the photovoltaic irradiation power in the four different seasons, i.e., spring, summer, autumn, and winter. The prediction accuracy was generally higher for 5 min using all three compared methods. And the accuracy gradually decreased for 10, 20, and 30 min time intervals. In all cases, the proposed hybrid deep learning model maintained better prediction performance compared to the other two methods. Moreover, the advantage became more obvious when the time interval expanded.

In summary, the prediction results show that the difference between the three models is small in the seasons with fewer fluctuations, e.g., in summer and winter. However, the proposed GRU_Attention model has better performance over all compared methods. It is evident that the model GRU_Attention proposed in this paper adapts to the diversity of solar irradiance changes in different seasons. In contrast, the direct applications LSTM and GRU do not fit the actual solar irradiance change curve well. It is evident that the proposed hybrid deep learning model is more stable and simultaneously improves the accuracy of solar irradiance prediction.

## 4. Conclusions and Future Works

This study proposes a hybrid deep learning solar irradiance prediction model, namely, GRU_attention, based on the Keras framework, which has the advantages of strong generalization ability, fast modeling, high portability, and high prediction accuracy. This study mainly introduces the principle and structure of the network, the overall framework, etc., and describes the specific structure of the entire end-to-end model. In particular, a GRU_attention convolutional neural network automatically extracts data features, overcomes the problem of high dimension of the original data, improves model learning performance, and provided accurate forecasting results for solar irradiance at the Nevada desert in the USA. Deep learning has excellent forecasting capabilities, especially for large and medium-sized data sets. Combining different deep learning techniques effectively improves the accuracy of power generation forecasting results by increasing the reliability and stability of these results. In the future, we will further expand the time interval to make medium-term to long-term solar irradiance predictions. Future research directions include exploring the PV system damage caused by the severe fluctuations of solar irradiance.

**Author Contributions:** Conceptualization, K.Y. and H.S.; methodology, K.Y.; software, H.S.; validation, H.Z., H.S. and K.Y.; formal analysis, L.W.; investigation, L.W.; resources, L.W.; data curation, M.X.; writing—original draft preparation, K.Y.; writing—review and editing, K.Y.; visualization, H.S.; supervision, K.Y. and Y.M.; project administration, K.Y. and Y.M.; funding acquisition, K.Y. and Y.M. All authors have read and agreed to the published version of the manuscript.

**Funding:** This work was supported by Zhejiang Provincial Natural Science Foundation of China under Grant No. LY19F020016, in part by the National Natural Science Foundation of China under Grant 61850410531 and 61602431 (K.Y.), in part by the research project on the "13th Five-Year Plan" of higher education reform in Zhejiang Province, under grant number JG20180526 (M.X.) and in part by the National Natural Science Foundation of China under Grant 61972156, and Program for Innovative Research Team in Science and Technology in Fujian Province University (Y.M.).

**Conflicts of Interest:** The authors declare no conflict of interest.

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
