# Peer review of "Short-Term Solar Irradiance Forecasting Based on a Hybrid Deep Learning Methodology"

_information, doi:10.3390/info11010032_

Round 1
Reviewer 1 Report
The paper presents a combined gated recurrent unit neural network with attention mecanism for the short-term energy generation prediction of a photovoltaic power plant. The obtained result are quite good. However, there are some aspects that are not totally clear in the paper:
What are the input parameters (time of day, solar irradiation, temperature, wind speed ...) of the implemented hybrid deep learning method? How big is the MIDC Las Vegas photovoltaic power plant? Do the implemented system predict the power generation of the entire PV farm or of some PV panel clusters. At "3.Results" section the authors mention that the study was devided in 1 min, 3 min, 5 min and 10 min time interval, however in Table 1 and Fig. 7 to 10 the results are presented for 5 min, 10 min, 20 min and 30 min time intervals. Is this a typing mistake or the first set of time intervals refer to data collection time scale while the second set of time intervals refer to the length of the time interval that is predicted? If the second assumption is true then Fig. 7 to 10 for which data collection time interval present the obtained results. It is assumed to get different prediction accuracy if different data collection time step is used. Fig. 7 to 10 are not clear enough, the axis labels could not be read. Also, due to similarity of the results obtained with different prediction techniques it is hard distinguish between the results provided by each prediction method. Please consider making bigger pictures or use different line styles to represent the obtained prediction results. Please specifice how the testing data sets were selected for the 4 implemented prediction system (one for each season: spring, summer, etc. ) Are there 4 seasons in Las Vegas region or only two "summer" and "winter"?Author Response
The paper presents a combined gated recurrent unit neural network with attention mechanism for the short-term energy generation prediction of a photovoltaic power plant. The obtained result are quite good. However, there are some aspects that are not totally clear in the paper:
What are the input parameters (time of day, solar irradiation, temperature, wind speed ...) of the implemented hybrid deep learning method?
Reply: The input parameters (except the PV power output) includes the 'averaged wind speed' and 'Peak wind speed'. According to the literature, e.g., [1] and [2], the wind speed greatly influences the cloud movement and consequently influences the solar irradiance received by the PV panels (Last paragraph of the Introduction Section and First paragraph of Section 2.4). Above description has been added to the revised version of the manuscript. All modifications made are marked in red in the revised version.
How big is the MIDC Las Vegas photovoltaic power plant?
Reply: Thanks for the reviewer's comments. We would like to clarify that the data was collected by University of Nevada @ Las Vegas and NREL to support solar power generation research in the states [3]. The measurement station is located at:
Latitude: 36.107o North
Longitude: 115.1425o West
The original data only contains the solar irradiance instead of the PV energy generation output. However, PV output is always proportional to the solar irradiance. As a result, in fact, we are not monitoring the actual PV plant in Nevada desert. Instead, we are predicting the solar irradiance in short-term for a particular location at Nevada desert. The prediction results can be helpful predicting PV energy generation output, since 1) solar irradiance is directly proportional to the energy output; and 2) there are many PV plants in Nevada desert. The above information has been added to the revised version of the manuscript (Last paragraph of the Introduction Section and First paragraph of Section 2.4). All modifications made are marked in red.
Do the implemented system predict the power generation of the entire PV farm or of some PV panel clusters.
Reply: Thanks for the reviewer's question. As explained in the last question, we are not predicting the PV output. Instead, we are predicting the solar irradiance in Nevada desert, which is proportional to PV output if there is a PV station located at the measurement point.
At "3.Results" section the authors mention that the study was devided in 1 min, 3 min, 5 min and 10 min time interval, however in Table 1 and Fig. 7 to 10 the results are presented for 5 min, 10 min, 20 min and 30 min time intervals. Is this a typing mistake or the first set of time intervals refer to data collection time scale while the second set of time intervals refer to the length of the time interval that is predicted? If the second assumption is true then Fig. 7 to 10 for which data collection time interval present the obtained results. It is assumed to get different prediction accuracy if different data collection time step is used.
Reply: The original statement in 3. Result is a typo. It has been corrected in the revised version (marked in red). In fact, we did several experiments and tested the prediction results for time intervals 1, 3, 5, 10, 20 and 30 mins. The paper has been changed from its original version, which is 1, 3, 5, 10 mins, to the latest version @ 5, 10 , 20 and 30 mins. Because we think that longer time interval forecasting is more meaningful.
Fig. 7 to 10 are not clear enough, the axis labels could not be read. Also, due to similarity of the results obtained with different prediction techniques it is hard distinguish between the results provided by each prediction method. Please consider making bigger pictures or use different line styles to represent the obtained prediction results.
Reply: Thanks for the reviewer’s comments. We have re-draw Figs 7-10 using thicker lines, higher resolution and different line style, i.e., using diamonds, triangle, square, etc., attaching to the lines. We have enlarged the Figures 7-10 and shown one subfigure for each season, i.e., spring, summer, autumn and winter.
Please specifice how the testing data sets were selected for the 4 implemented prediction system (one for each season: spring, summer, etc. ) Are there 4 seasons in Las Vegas region or only two "summer" and "winter"?
Reply: Thanks for the reviewer’s nice comments. We agree that we did not make it very clear how we divided the dataset into training and testing sets. In the revised version, we rephrased the first paragraph of Section 2.4: “The data of the whole year 2014 was used. For each season, one month of data was selected as testing dataset and the remaining two months were treated as training dataset. As a result, we have eight months’ solar irradiance data for training and four months’ data for testing.” Modifications made are marked in red in the manuscript.
Reference:
[1] Ssekulima, E. B., Anwar, M. B., Al Hinai, A., & El Moursi, M. S. (2016). Wind speed and solar irradiance forecasting techniques for enhanced renewable energy integration with the grid: a review. IET Renewable Power Generation, 10(7), 885-989.
[2] Chen, X., Du, Y., Lim, E., Wen, H., & Jiang, L. (2019). Sensor network based PV power nowcasting with spatio-temporal preselection for grid-friendly control. Applied Energy, 255, 113760.
[3] Stoffel, T., & Andreas, A. (2006). University of Nevada (UNLV): Las Vegas, Nevada (Data) (No. NREL/DA-5500-56509). National Renewable Energy Lab.(NREL), Golden, CO (United States).

Reviewer 2 Report
In my opinion, the manuscript titled "Short-Term Solar Photovoltaic Energy Generation Forecasting based on Hybrid Deep Learning Methodology can be published in INFORMATION.
Authors proposed the hybrid deep learning framework to forecasting the energy generation changes of the solar photovoltaic system. The research improved the prediction accuracy of the output power of photovoltaic systems and provided a basis for the proposed new hybrid model.
In my opinion, Authors should expand Introduction.
The author should describe the obtained results in more detail.
Details of my comments are in the attached file.

Author Response
In my opinion, the manuscript titled "Short-Term Solar Photovoltaic Energy Generation Forecasting based on Hybrid Deep Learning Methodology can be published in INFORMATION.
Authors proposed the hybrid deep learning framework to forecasting the energy generation changes of the solar photovoltaic system. The research improved the prediction accuracy of the output power of photovoltaic systems and provided a basis for the proposed new hybrid model.
In my opinion, Authors should expand Introduction.
Reply: Thanks for the reviewer’s comments. The introduction has been expanded. And a lot more details of the proposed method have been added. It is noted that all modifications made are marked in red in the revised version.
The author should describe the obtained results in more detail.
Reply: We have revised the results discussion part and added one more paragraph describing the obtained results (Second last paragraph of Section 3). Modifications made are marked in red.
Details of my comments are in the attached file.
Reply: We have made modification point-by-point according to the reviewer’s comments. All modifications made are marked in red.

Round 2
Reviewer 2 Report
In my opinion, the article entitled "Short-Term Solar Irradiance Forecasting based on Hybrid Deep Learning Methodology" in the present form can be published in INFORMATION.